# Efficacy of a New Low-Protein Multimedia Diet App for PKU

**DOI:** 10.3390/nu14112182

**Published:** 2022-05-24

**Authors:** Sharon Evans, Catherine Ashmore, Anne Daly, Perninder Dhadwar, Atif Syed, Olivia Lecocq, Richard Jackson, Alex Pinto, Anita MacDonald

**Affiliations:** 1Birmingham Women’s and Children’s Hospital NHS Foundation Trust, Birmingham B4 6NH, UK; catherine.ashmore@nhs.net (C.A.); a.daly3@nhs.net (A.D.); olivia.lecocq1@nhs.net (O.L.); alex.pinto@nhs.net (A.P.); anita.macdonald@nhs.net (A.M.); 2Imobisoft, The Technocenter, Coventry University Technology Park Puma Way, Coventry CV1 2TT, UK; perninder@imobisoft.co.uk (P.D.); atif@imobisoft.co.uk (A.S.); 3Liverpool Clinical Trials Centre, University of Liverpool, Brownlow Hill, Liverpool L69 3GL, UK; r.j.jackson@liverpool.ac.uk

**Keywords:** phenylketonuria (PKU), app, multimedia, low-protein diet

## Abstract

Patients with phenylketonuria (PKU) require a phenylalanine/protein-restricted diet, with limited food choice. Interpreting food labels, calculating protein intake, and determining food suitability are complex and confusing tasks. A mobile multi-media low-protein diet app was developed to guide food choice, label interpretation, and protein calculation. ‘*PKU Bite*’^®^ includes >1100 specialist and regular low-protein foods, is colour-coded for suitability, and features a protein calculator. A 12-week randomised controlled trial assessed app efficacy, compared with written/pictorial material, in 60 parents/caregivers of children with PKU, aged 1–16 years, and 21 adolescents with PKU. Questionnaires examined self-efficacy and label-reading knowledge; food records evaluated natural-protein intake, compared with prescriptions. There was no difference between groups in label-reading knowledge or self-efficacy, but there was a trend for improved accuracy of dietary protein calculation, when using the app (baseline/12-weeks: app 35%/48%; control 39%/35%). Parents of children <10 years of age (median 5.5 years), were most likely to use the app to check the phenylalanine/protein content of a food or to verify suitability of foods. Whilst the app was popular (43%), so too was contacting the dietitian (43%), using written/pictorial information (24%), or using social media (18%). This is the first dietary app for PKU to be studied in a systematic way as well as validated by healthcare professionals. It is a useful adjunct to existing resources and will be a valuable tool for educating parents of younger children.

## 1. Introduction

Stringent dietary phenylalanine (Phe) restriction is an essential treatment strategy, to prevent severe neurological sequelae in children with phenylketonuria (PKU). This inborn error of metabolism is characterised by a deficiency of the enzyme phenylalanine hydroxylase, essential for the metabolism of the amino acid phenylalanine, leading to accumulation of Phe in the blood and brain. Phe is found in protein-containing foods, and protein intake is commonly reduced to <10 g/day (<500 mg Phe) in patients with classical PKU, to maintain metabolic control [1].

The amount of natural protein tolerated in PKU is dependent on the PKU phenotype. In the UK, protein is allocated in the form of an ‘exchange’ system (one exchange is the amount of food that provides one gram of protein, or 50 mg Phe for fruit and vegetables). This provides some flexibility in food choice. High-protein foods (e.g., meat, fish, eggs, cheese) and the artificial sweetener aspartame (high in Phe) are avoided, and plant-protein food sources are calculated accurately, to maintain satisfactory blood Phe control. Remaining protein requirements are provided by Phe-free L-amino acids or glycomacropeptide (a low Phe peptide), which is supplemented with added amino acids. Energy is supplied by specially manufactured low-protein foods such as bread, pasta, and cereals [1], as well as regular foods that are naturally low in protein.

A Phe-restricted diet is complex, confusing, and, even, overwhelming. For example, whilst most foods contain 50 mg of Phe for each gram of protein, fruits and vegetables vary in their Phe content, and no Phe analysis is available for regular manufactured foods, although protein content is reported on food labels. European legislation states that manufacturers can declare foods as protein free, if they contain a threshold of ≤0.5 g/100 g [2]. Thereby, a food product may have a protein-containing ingredient listed, but the nutrition label states 0 g/100 g protein [3,4]. Ingredient lists on manufactured products can, also, include an extensive range of unusual additives or food ingredients. Novel and exotic foods and ingredients are continually introduced. Recently spirulina, a type of algae, has been used as an additive in sweets and smoothies to give a green colour, but it is high in protein. Powdered custards, dessert mixes, drinking chocolate, and milkshake powders regularly describe nutrient values, assuming they have been reconstituted with milk or egg, which are not permitted in a Phe-restricted diet; so, it is challenging to identify the protein content of the powder only [3]. Initiatives to lower the sugar content of manufactured foods have led to a higher use of aspartame [5]. In addition, some ingredients (e.g., spice mixes) used in small quantities may have high protein content, so uncertainty exists about their suitability. A recent survey of individuals with PKU or their caregivers reported that 31% found it difficult to calculate food protein exchanges from food labels, and 90% had experienced problems with food labelling in the previous six months [3].

There is substantial patient/caregiver demand for quick access to information about the suitability of foods for PKU. On social media (Facebook/Instagram), caregivers/patients discuss interpretation of protein on food labels and suitability for inclusion in a low-protein diet. Whilst this is a useful forum, with input from health professionals, sometimes inaccurate information may be shared and poor dietary practice perpetuated. Furthermore, professional written/pictorial information may be misplaced or inaccessible when required, prompting phone calls to dietitians for quick advice on individual food products and the suitability of a food.

Mobile media are commonly used to access dietary information, but current apps for PKU are often commercially developed, country specific, under researched, inattentive to food suitability, or not compatible with all devices [6]. No existing UK apps provide information on food suitability for PKU, and none have been validated by a healthcare organisation or certified by a recognised body. However, studies in other health and medical conditions have demonstrated that apps can improve adherence compared with traditional interventions, such as written educational information [7,8,9,10,11,12,13,14,15,16,17].

We have developed a UK mobile app, specifically written for PKU, accessible on iOS, Android, and web-based smart phones and tablets. The app contains a databank of 1100 specialist and regular low-protein foods/ingredients that might be consumed on a Phe-restricted diet. It was designed to inform patient/caregivers about food labelling interpretation, give information about food exchanges (protein/Phe), aid calculation of food protein exchanges, and delineate the suitability of each food item. The principles of the dietary information within the app are based on professional consensus, by UK Inherited Metabolic Disorder (IMD) dietitians utilising Delphi methodology [18,19].

The aim of this study was to investigate if the use of a mobile app in PKU was able to increase self-efficacy in dietary management, improve knowledge and interpretation of food labels, and improve the accuracy of patient/caregiver daily food protein calculations, when selecting fresh, manufactured, and specialised dietary products in a low-protein diet, compared with use of ‘traditional’ written dietary resources.

## 2. Materials and Methods

This was a 12-week randomised, controlled, parallel, intervention study. It aimed to determine if the use of a mobile application (available via any smart phone or tablet) by caregivers of, and patients with PKU, who are following a strict low-protein diet, improved food label interpretation, knowledge of food suitability for inclusion in the diet, protein calculations, dietary adherence, and self-care, compared with control subjects who used written/pictorial information only. All subjects from both the app and control groups were then encouraged to download and use the app, and were followed up for a further six months. Figure 1 shows the app study design and procedures.

### 2.1. App Development

#### 2.1.1. App Content

A multimedia app, specifically for PKU, was developed by a team of 5 specialist IMD dietitians, in consultation with the software development company Imobisoft. App content for over 1100 specialist and regular low-protein foods/ingredients was developed over a 2-year period.

This included for every food:a general description of the food such as appearance and origins;suitability for inclusion in a low-protein diet (based on British Inherited Metabolic Disease Group (BIMDG) UK dietetic consensus statements [18,19], including colour coding (**green:** low-protein exchange-free foods; **orange:** exchange foods that contain protein/Phe and so should be calculated as part of the daily allowance; **red:** high-protein foods that are best avoided);menu ideas and food serving suggestions;exchange-free (low-protein) recipes;food preparation;food storage recommendations;food availability information, including details of home delivery companies that supply low-protein prescription items.

Foods were selected for their suitability for a low-protein diet, including specialist low-protein foods, foods naturally low in protein, and foods that are calculated as part of the daily protein allowance. Some high-protein foods were included with a warning that they were best avoided. All app data were cross-checked individually, by 3 IMD dietitians at least twice, and by 2 dietitians at least 5 times. Users could search by food name or food category. Food categories (*n* = 12) included:all special low-protein prescription items;plant foods (e.g., fruits and vegetables, whether fresh, frozen, canned, or dried);cheese, yogurt, cream, and alternatives (e.g., regular, low-protein, or vegan cheeses, yoghurts, cream);savoury foods/meat alternatives (e.g., low-protein or plant burgers, sausages, soups, legumes, plant-based meat alternatives);flours, pasta, and cereals (e.g., breads, bread products, and pastries, including gluten-free varieties);fats, oils, sauces, dips, and spices (e.g., butter, margarine, oils, gravy, ketchup, dressings, marinades);cakes, biscuits, and desserts (e.g., plant and milk-based ice-cream, custards, jellies, puddings);drinks (e.g., fizzy drinks, squash, juices, smoothies, milk, plant milks, hot chocolate, milkshake powders);baking ingredients (e.g., sugar, baking powder, cake decorations, colourings, flavourings);sweet spreads, sweets, and syrups (e.g., chocolate, sweets, chewing gum, jams, honey, chocolate spread);snack foods (e.g., crisps, crackers, ice lollies);food label ingredients (e.g., food additives).

The app also included a protein exchange calculator, for calculating the protein content of a food per 100 g or per portion, using information from food labels.

Any foods, food groups, or ingredients that may contain aspartame were identified with a red warning triangle with an exclamation mark and the words “aspartame warning”. In addition, there was a filter function to enable all possible aspartame-containing foods/products in the app to be listed.

The app was written in the English language only.

#### 2.1.2. App Design

The app was designed and developed over 4 months, which included choice of name (‘*PKU Bite*’^®^), logo design (Figure 2), background, and page designs. During this development phase, the app design was discussed with user focus groups, including parents of children with PKU (*n* = 7) and children with PKU (*n* = 4), to collect feedback on usability and to direct development.

### 2.2. Randomised Controlled Trial

#### 2.2.1. Subjects

Eighty-one patients aged 10–16 years, or caregivers of children aged 1–16 years, on low-protein diets for PKU were recruited (40 study and 41 control subjects). Subjects were recruited from one specialist IMD centre (Birmingham Women’s and Children’s NHS Foundation Trust). Subjects were excluded, if the child with PKU had co-morbidities (e.g., cancer, inflammatory bowel disease, diabetes), or was following a special diet for any other medical reason. All subjects were well established on their Phe-restricted diet.

#### 2.2.2. Randomisation

Individual subjects or families were randomised, by computer-generated random number block sequences (blocks of 2 and 4), to use the app, called ‘*PKU Bite*’^®^ (study group), or to use written/pictorial resources (control group) only.

#### 2.2.3. App Group

Study group subjects were sent a trial download of the ‘*PKU Bite*’^®^ app for their phone/tablet and attended a hands-on 30 min training session, to explain how to download the trial app and use the app functions. A copy of the teaching slides was given for reference. Subjects were asked to use the app for 12 weeks, as a source of information for food suitability, and to use the exchange calculator to determine the appropriate portion size for any protein-containing foods searched. They could use the recipes, menu ideas, preparation, storage, and availability information, as required. Other non-app sources of dietary information available to this group included: direct contact with an IMD dietitian, any previously distributed written/pictorial information, social media (e.g., Facebook/Instagram), and the NSPKU (National Society for PKU) webpage.

#### 2.2.4. Control Group

Control group subjects were issued with an NSPKU exchange calculator card, dietary information book, and written pictorial product information booklets, produced by the IMD dietetic team. This included pictorial guides on: the basic PKU diet; food exchanges; low-protein fruits and vegetables; ice lollies, ice-creams, and sorbets; low-protein cheese; low-protein baking and desserts; plant milks, creams, yoghurts, and desserts; low-protein sweets; and savoury low-protein alternatives. This information was comparable to the data on the app. They, also, had direct access to an IMD dietitian and information from social media. They used these resources as a reference for 12 weeks for information on food suitability, protein exchange amounts, and menu ideas.

#### 2.2.5. Demographic Questionnaire

At baseline, all subjects completed a demographic questionnaire, describing parent/child age, gender, ethnicity, parent educational level, frequency of use of social media, any apps currently used, and other dietary resources.

#### 2.2.6. Knowledge Questionnaire

A 22-item multiple choice non-validated questionnaire on low-protein labelling knowledge and interpretation (see Appendix A) was completed, by all subjects at baseline and after 12 weeks of using the app (app group) or written educational material (control group). The questionnaire was pilot tested with a user group of adolescent patients with PKU or their parents and IMD dietitians, prior to the study. It included pictures of 22 manufactured foods, information on their protein content, and questions about whether these products could be included in a Phe-restricted diet. For example, how many 1 g protein exchanges would a specified weight of a food product contain, or how much of a food product could be eaten for 1 exchange, when the protein content per 100 g was given. Information given on the app or in the control written information supported subjects in answering these questions.

#### 2.2.7. Self-Efficacy Questionnaire

This validated 8-item questionnaire measured how confident subjects were with managing different aspects of their own, or their child’s PKU health care, on a 10-point Likert scale (not confident = 1, to totally confident = 10) [20]. This included questions such as “I can choose appropriate foods [for my child] to eat when hungry” and “I know what to do when [my/my child’s] blood Phe level goes higher or lower than it should be”. This was completed by all subjects at baseline and at 12 weeks.

#### 2.2.8. Patient Activation Measure

This validated 13-item questionnaire, with a 4-point Likert scale (strongly disagree to strongly agree) [21], was completed by all subjects at baseline and at 12 weeks. Similar to the self-efficacy questionnaire, this also measured confidence with PKU self/child-care management. Examples of questions included: “I am confident I can take action to minimise symptoms/problems associated with [my/my child’s] PKU”, “I am confident I can maintain lifestyle changes like diet even during times of stress”, and “I understand the nature and cause of PKU”.

#### 2.2.9. Feedback Questionnaire

A non-validated feedback questionnaire on subject frequency and reasons for use of the app, written information, and other resources used during the study period was completed at 12 weeks.

#### 2.2.10. Natural Protein Intake

A 24 h dietary recall was completed at baseline, 6 weeks, and 12 weeks. Subjects (caregivers and patients) were asked to record the foods they calculated/measured, as part of their protein prescription, and the number of food exchanges (1 g protein/50 mg Phe) allocated for each item. An IMD dietitian then checked the number of food exchanges actually eaten, compared with prescribed amounts, and documented any calculation errors.

#### 2.2.11. Metabolic Control

All routine weekly blood Phe results were recorded, for 6 weeks prior to study commencement and throughout the study period. Mean blood Phe results were then calculated for each child at baseline, 12 weeks, and 6 months.

### 2.3. Six Month Follow Up

At the end of the 12-week randomised controlled trial (RCT), control subjects were sent a download of the ‘*PKU Bite*’^®^ app for their phone/tablet, and they attended a 30 min training session on the app. Both groups (app and control), then, used the app for a further 6 months, completing the self-efficacy, patient activation measure, and feedback questionnaires on completion. Routine weekly blood Phe results were, also, recorded.

### 2.4. Statistics

From a hospital patient population of approximately 120 children with PKU, a sample size of 80 (40 study, 40 controls) had 80% power to detect a 30% improvement in test scores for protein calculation in the study group (using the app), compared with the control group (not using the app), with a significance level of 0.05 (two-tailed). Continuous data are summarised as median (IQR), and categorical data are summarised as frequencies of counts, with associated percentages. Comparisons of outcome data between treatment groups or timepoints were performed using Wilcox tests.

### 2.5. Ethical Approval

This study was conducted according to the guidelines laid down in the Declaration of Helsinki, and a favourable ethical opinion was obtained from the West Midlands–Edgbaston National Research Ethics Service (NRES) Committee (REC reference: 20/WM/0010 and IRAS ID: 260370). Written informed consent was obtained from the parent/carer of all children, as was assent from the children where appropriate, according to level of understanding.

## 3. Results

### 3.1. Subjects

Table 1 describes participant characteristics. There was no significant difference between the demographic characteristics of the two groups, except for control group mothers, who had a lower median level of education compared with the app group mothers (2.0 vs. 3.0; *p* = 0.04 Mann–Whitney). Six of the control group parents/carers (all mothers) and three of the children (two female) declined to do the app training, so did not complete the final six months using the app. Reasons for withdrawal included: disengagement with technology +/− adolescent fixed-eating habits (*n* = 3); patient transfer to an adult hospital (*n* = 2); extended holiday (*n* = 1); family social issues (*n* = 1); and patient psychological/psychiatric issues (*n* = 2). Sensitivity analyses, which restrict the patient population only to those patients who remained in the study, were performed. These had no meaningful impact on any of the study’s interpretations (data not shown).

### 3.2. Baseline Use of Apps and Sources of Dietary Information

The top five apps, used by more than 50% of the children, were YouTube, TikTok, Snapchat, WhatsApp, and Instagram, whilst for parents/carers they were WhatsApp, Facebook, Twitter, YouTube, and Messenger. Parents used more apps than children. There was no significant difference between the groups for the total number of apps used, overall, weekly, or daily (Table 2). Parents/carers used a median of 36 different apps, 10 weekly and 5 daily; whilst children used a median of 30 apps, 7 weekly and 4 daily. However, there was wide variation in app use within the groups.

For parents/carers at baseline, written/pictorial dietary information, contacting the dietitian by phone or email, and social media (Twitter/Facebook) were the most common (>50% of subjects) sources of low-protein diet information (Table 2). For children, the most common source was the dietitian, followed by written/pictorial information, and nearly one quarter reported that they relied on their parent(s). There was no significant difference between parent/carer or child app and control groups, for the source(s) of low-protein dietary information used at baseline.

### 3.3. Frequency of Using the ‘PKU Bite’^®^ App at Twelve Weeks and Six Months Follow Up

The app was used at least once a week, by 58% (*n* = 11) of female and 40% (*n* = 4) of male parent/carers in the app group at 12 weeks, and 53% (*n* = 10) of females and 60% (*n* = 6) of males at 6 months (Table 3). In the control group, at 6 months (after using the app) 46% (*n* = 12) of females and 60% (*n* = 3) of males used the app at least weekly. Amongst children, only a few used the app regularly: at 12 weeks, in the app group, 60% (*n* = 3) of male and 17% (*n* = 1) of female children used the app at least weekly, but by six months only one male and one female used it weekly, whilst two female control children used the app weekly.

When the parents/carers were divided into app users (used the app at least once a week at six months; *n* = 31) and app non-users (rarely/never used the app at six months; *n* = 29), there was a significant difference in the age of their children (median age (IQR): users 5.5 years (1.3–10) vs. non users 10.0 years (6–13); *p* = 0.002 Wilcoxon signed rank), suggesting that the app was more likely to be used by parents/carers with children under the age of 10 years.

### 3.4. Frequency of Seeking Dietary Assistance at Baseline, Twelve Weeks, and Six Months

#### 3.4.1. Between Group Differences

At baseline, more control group parents/carers reported using written dietary information at least weekly than the app group (carers: 71% vs. 34%), but this was not statistically significant at 12 weeks or 6 months (Table 3). More control children reported contacting the dietitian at least weekly than the app group at baseline (50% vs. 9%) and at 12 weeks (30% vs. 0%). 

#### 3.4.2. Within Group Changes

During the study, there were some reported changes in the frequency of dietary resources used within groups (Table 4). Fewer app group parent/carers used social media at least weekly at 12 weeks and 6 months, compared to baseline (41% vs. 72%; *p* = 0.0003; 52% vs. 72%; *p* = 0.01). Fewer control group parent/carers reported contacting the dietitian and using social media and written information at least weekly at six months than at baseline (dietitian: 28% vs. 42%; *p* = 0.06) (social media: 40% vs. 65%; *p* = 0.001) (written information: 52% vs. 71%; *p* = 0.0009). 

There were no statistically significant changes for either app or control group children during the study, in the reported frequency of using any dietary resources, however, subject numbers were small.

### 3.5. Self Confidence in Managing PKU

Throughout the study, there were only small changes in self-confidence with managing PKU for parents/carers or children in either group, when measured by self-efficacy or the patient-activation measure (Table 5).

Self-efficacy: children in the app group had more self-efficacy than controls after six months (median score 8.20 vs. 6.65; *p* = 0.01), but this was, also, the case at baseline (7.95 vs. 6.35; *p* = 0.02), suggesting there was no real change (Table 5). App group children were less self-confident at 12 weeks compared to baseline, but improved after 6 months on the app, although results were not significant. Similarly, app group carers had no significant change in self-confidence. However, whilst app group carers were more confident at baseline than controls (median 8.85 vs. 8.20; *p* = 0.02), at six months control carers improved significantly compared with baseline (8.20 vs. 9.20; *p* = 0.02), suggesting control carers had improved self-efficacy. Whilst the six carers who did not complete the six-month questionnaires may have skewed the baseline data (based on qualification levels of 0–3 [22]), when their data was omitted, results did not differ. When all parents/carers (from both app and control groups) were divided into users (used the app at least once a week at six months; *n* = 31) and non-users (rarely/never used the app at six months; *n* = 29), there was no significant difference in self-efficacy scores at baseline and six months, either within or between groups.

Patient activation measure: control group carers were more self-confident after six months of using the app, compared with baseline (carers: mean score 3.60 vs. 3.70; *p* = 0.009) (Table 5). Similarly, compared with baseline, carers in the app group showed improved confidence after 12 weeks of using the app and after 6 months (3.70 vs. 3.60; *p* = 0.02), but there was no significant change for children. However, as with the self-efficacy questionnaire, app group children were significantly more confident at six months compared with controls (3.20 vs. 2.90; *p* < 0.0001). Control children were also less confident at six months than at baseline (3.10 vs. 2.90; *p* = 0.01). Patient-activation scores for app users versus non-users showed no significant difference between groups at baseline and six months, but app users did improve from baseline to six months (3.5 vs. 3.8; *p* = 0.03).

### 3.6. Knowledge and Interpretation of Low-Protein Labelling

There was no significant difference between or within groups for knowledge and interpretation of low-protein labelling, at baseline or 12 weeks, for children or carers (Table 6). When the app group parents/carers were divided into those who used the app regularly (≥weekly) at 12 weeks (*n* = 16) and those who rarely or never used the app (*n* = 13) at 12 weeks, there was a non-significant trend toward improved knowledge between baseline and 12 weeks for users, with higher scores than nonusers.

### 3.7. Accuracy of Calculating Daily Natural Protein Intake

There was a trend toward improvement in ability to calculate protein intake, for those subjects using the app. Both parents/carers and children using the app made fewer calculation errors at 6 and 12 weeks compared to baseline, and at 12 weeks there were more subjects making no errors (Table 7). In the control group, there was some improvement at 12 weeks, but not at 6 weeks for the parents/carers, and there was no change in errors throughout the 12-week study period for the control children. However, no differences reached statistical significance. When the app group was divided into users (used the app at least once a week at 12 weeks; *n* = 16) and nonusers (rarely/never used the app at 12 weeks; *n* = 13), there were no significant differences in the percentage of incorrectly calculated items, but both groups improved from baseline to 12 weeks, and both groups had more subjects correctly calculating all of their protein exchanges.

### 3.8. Metabolic Control

There was no significant difference between or within the app and control groups, for mean blood Phe levels at baseline, 12 weeks, or 6 months (Table 8), and all median levels were within the recommended target range of 120–360 µmol/L (<12 years) or 120–600 µmol/L (>12 years), indicating that the children had acceptable blood Phe control. However, there was a trend toward lower mean blood Phe levels at 12 weeks compared to baseline, in both groups, possibly associated with increased attention to diet in the initial weeks of the study.

### 3.9. Participant Feedback

Responses about the most useful features of the app are given in Table 9. Seeking advice on food suitability was a common use for the app, as well as looking up the protein/Phe content of a food, or the number of exchanges in a portion or per 100 g of a food. It was also noted to be helpful for accessing diet information quickly.

When subjects were asked for their preferred method of obtaining diet information, whilst the app was popular (43%) across all subjects, so too was contacting the dietitian (43%), followed by written information (24%) and social media (18%). Children were less likely than carers to use any of the methods, but contacting the dietitian was rated the highest (Table 10). Some respondents used a combination of different resources.

At the end of the study, subjects were asked how satisfied they were with the app. Overall, 51% percent (*n* = 37) were very or extremely satisfied, 21% (*n* = 15) were satisfied, and 8% (*n* = 6) were unsatisfied or very unsatisfied (19%, *n* = 14, did not respond).

There was a lot of positive feedback about the app and the functions that people found particularly helpful, such as the calculator and the mobile nature of the app (Table 11). Comments on what did not work well mostly related to being unable to find a specific food, a desire for more branded and individual food items, or difficulties with the calculator or technology in general. Some teenagers or their carers reported being ‘set in’ their dietary habits, so did not vary their diet much, meaning the app was not required.

When subjects were asked about additional app functions for the future, common themes were: an exchange tracker, a barcode/label scanner, more recipes and menu ideas, and a bigger range of branded foods (Table 11).

## 4. Discussion

This is the first dietary app designed specifically for PKU, which has been formally evaluated to assess efficacy. Feedback from the RCT and patient focus groups has facilitated development of a user-friendly app, with information explained in a clear effective way, and its value analysed objectively. Previous dietary apps for inherited metabolic disorders, both within and outside the UK, have not included the full range of criteria represented in this app, in terms of content, availability on all devices, non-commercial bias, and validation by health professionals or credible organisations [6].

The results of this RCT provide valuable insight into how this app is used by both adolescents with PKU and parents/carers of children with PKU, the key features used, and suggestions on how the app could be improved. It was apparent that whilst the app was popular, it could not replace other commonly used resources for obtaining dietary advice, such as contact with the dietitian, written/pictorial information booklets, and social media. There is substantial variation in individual preference, for methods of seeking dietary information; the app offers an additional choice to existing resources, for those who commonly use this form of technology.

During the time subjects used this app, there were trends toward improvement in the accuracy of protein calculation, self-confidence in dietary management, and reduced reliance on other dietary information resources. However, it is difficult to isolate this to solely the influence of the app, as subjects had access, concurrently, to other sources of dietary information, such as the dietitian, written/pictorial information, and social media. Both app users and nonusers improved in their protein calculations from baseline to 12 weeks, suggesting that the extra attention associated with diet during the study led to subjects being more vigilant with food calculations, irrespective of their source of dietary information. There was no evidence of improved labelling interpretation or metabolic control, during the study period. However, this study included subjects who were well established with and knowledgeable about the Phe-restricted diet, were already self-confident in management of their PKU care, and had good metabolic control. In particular, adolescents and their parents/carers had been accustomed to dealing with the Phe-restricted diet in excess of 10 years, and commonly self-reported that they were ‘set in their ways’ and they varied their diet very little. Therefore, they were not regularly seeking dietary information, and, when they did, they often preferred to use the method they were accustomed to, such as contacting the dietitian or using written/pictorial educational material. Initial interest in the app may have faded, when adolescents realised that it did not provide any further information than they already knew or could access using existing resources. As children with PKU move into adolescence, they typically exhibit decreasing dietary adherence, and, consequently, reduced metabolic control [23], which is associated with the adolescent perception of invincibility and a lack of acknowledgment that their health may be affected by their actions [24]. Engaging adolescents in education about their dietary treatment is challenging, and PKU is no exception. Teenagers are high-level users of smartphones and social media, so they would, generally, be expected to be more receptive to these platforms [25]. However, there is also evidence that apps that enable content creation, sharing, and networking are more effective in this age group [25]. This is supported by the top five apps used by adolescents in this study being largely social media apps. Hence, an information-giving app such as ‘*PKU Bite*’^®^ is less likely to engage adolescents. Interactive features, such as a diet or exchange tracker or an ability to share recipes, might generate higher engagement.

At the end of the study, 50% of subjects were using the app regularly (at least weekly), whilst a quarter had rarely used it, and a quarter not at all. Uptake by adolescents was particularly poor, with only 4 out of 21 using the app regularly. In addition, it tended to be the parents of younger children that used the app frequently, rather than those with teenage children. For those not using, or rarely using the app, no change in knowledge, accuracy of protein calculation, or self-efficacy would have been expected. This lack of uptake is similar to other studies looking at educational dietary resources and reinforces the concept that provision of the resource alone does not guarantee motivation to utilise [26]. Furthermore, there is much evidence to suggest that a combination of teaching methods incorporating different learning styles is more effective than one resource or intervention alone, so expecting a solitary resource to engage all people with PKU is an unrealistic target [24,26,27,28].

Even with patient/caregiver education and provision of written/pictorial material, patients/caregivers find reading and interpretation of food labels confusing, leading to errors in protein calculation [3]. In addition, patients/caregivers commonly interpret suitability of foods differently to professionals, and, even within the same family, caregivers can vary in the interpretation of food-protein values on food labels. A study of 45 patients/caregivers of children with PKU and 49 dietitians demonstrated that dietitians are generally more relaxed about protein labelling; patients/caregivers were more likely to use exact protein analysis than dietitians (38%vs. 6%) and calculate the Phe content of special low-protein foods (64% vs. 30%) [29]. Uncertainty about interpreting food labels may, therefore, cause over-restriction in a diet that is already very limited. The app provides concordance in label interpretation, and targets an unmet national need to deliver more consistent care for patients with PKU who are on strict low-protein diets. Used in conjunction with other forms of dietary information, it provides another tool with the potential to impact and improve longer-term clinical outcomes.

The ‘*PKU Bite*’^®^ app provides caregivers/patients with accurate, reliable, and readily available information about the protein content of foods, with the aim of improving self-care skills in the interpretation of food-protein suitability. It is free to download from the App Store and Google Play and includes a brief set of instructions on how to use the app, so it is accessible for all patients with PKU in the UK. It is estimated that approximately 2360 patients are currently monitored by NHS hospitals in England [30]. It can also be used by parents/carers, extended family, friends, occasional carers, manufacturers of dietary products, catering companies, and school and nursery workers. It may also be useful for adult patients with PKU returning to a diet and for maternal PKU patients as a tool to re-educate. It is hoped that widespread availability will assist with UK standardisation of dietary instruction and minimise conflicting information.

Use of dietary resources, such as ‘*PKU Bite*’^®^, has the potential to reduce reliance on dietitians for basic dietary queries, thereby enabling more time for complex clinical care. It was originally intended that any contact made by subjects with IMD dietitians with dietary queries during the initial 12-week study period, and the 6-month follow-up period, would be recorded, to see if the app helped with reducing dietetic contacts. However, this proved to be too complex, due to the multi-faceted nature of phone calls. Parents/carers frequently call for other reasons, but will commonly use the opportunity to ask about suitability of different foods. Similarly, during dietetic calls to report blood Phe results, questions about food suitability commonly arise. In addition, the app often identified new foods they could eat, prompting clarification with the dietitian. Thereby, ‘*PKU Bite*’^®^ may have initially increased dietetic contacts, but for appropriate reasons.

There was some evidence that use of social media and written information had decreased during the six-month study period. However, it is difficult to isolate this as being due solely to the introduction of the app. Choice of dietary resources is influenced by many factors, including individual preference and learning styles. For PKU, educational material is usually issued on diagnosis and new materials are issued at regular intervals. Therefore, these documents were already available to all subjects, however, they may have been misplaced, lost, or forgotten. In addition, older children, and parents of teenagers, are more familiar with the dietary restrictions and may refer less often to informational resources, but the release of new booklets may generate interest in new products, thereby altering their normal eating behaviour. Issuing written/pictorial educational material and using the app, may also have drawn additional attention to the diet, potentially leading to changes in dietary patterns and behaviours, such as protein calculation.

Our intention is to further develop the ‘*PKU Bite*’^®^ app and add functions specifically requested by users, including a label scanner to calculate protein content of manufactured foods, a diet and formula tracker for users to record and monitor daily protein intake, and a pre-clinic assessment questionnaire to assist with collecting data for clinic appointments. This will include a validated food-frequency questionnaire that can also be used for research purposes [31]. In addition, a DTAC application (Digital Technology Assessment Criteria) is being submitted to NHSX—a UK government organisation responsible for setting national policy and developing best practice for National Health Service technology, digital information, and data, including data-sharing transparency [32]. This assessment is the national baseline criteria for digital health technologies entering into the NHS and social care [33]. It gives users confidence that the app meets clinical safety, data protection, technical security, interoperability, usability, and accessibility standards [33]. In the future, the app could be adapted to suit the dietary requirements of other inborn errors of amino acid metabolism.

There were some study limitations. Six mothers and three children in the control group chose not to complete the final six months of the study, so they did not download the app or attend the training session. Consequently, there was no six-month questionnaire data for nine control subjects. This included subjects with lower education, which may have influenced their engagement. However, when sensitivity analyses, which restrict the patient population only to those patients who remained in the study, were performed, these had no meaningful impact on any of the study’s interpretations. These were individuals who, whilst able to commit to the completion of questionnaires as control subjects, were less confident about committing to using the app. Seven of the nine subjects who withdrew from the study were adolescents, or parents of adolescent children, who were more likely to be set in their ways and resistant to change. The two subjects with younger children had family commitments and social circumstances that made it difficult to commit. It is likely that there were similar subjects in the app group who, whilst remaining on the study, reported not using the app. Randomisation of subjects in the study was complicated by families where more than one member participated (e.g., parent and child). In order to mimimise confounding variables, it was necessary to ensure that all members of the same family were randomised to the same study group. This may have had a bearing on the lack of homogeneity in the education level of mothers across study groups and the number of withdrawals from the control group, due to disengagement and social issues. Follow-up multicentre studies on the longer-term use of ‘*PKU Bite*’^®^ and its potential impact on dietary adherence, metabolic control, and reliance on dietitians for diet queries would be beneficial. It is, particularly, important to examine its efficacy alongside differing teaching styles and care protocols. Moreover, focusing on a cohort of parents of younger children, who are still learning about PKU, may be helpful.

## 5. Conclusions

‘*PKU Bite*’^®^ is a dietary app, specifically designed for PKU, which has been developed in close partnership with patients and caregivers, independent of industry involvement, and has been extensively studied to examine its efficacy. The app has the potential to improve protein calculation, knowledge, and self-efficacy with PKU management, and is one of a multitude of tools that can be used to educate and inform patients and parents/carers of children with PKU. Validation by a UK national approval process, supported by the NHS, will further support its use as an adjunct to existing resources for PKU.

## Figures and Tables

**Figure 1 nutrients-14-02182-f001:**
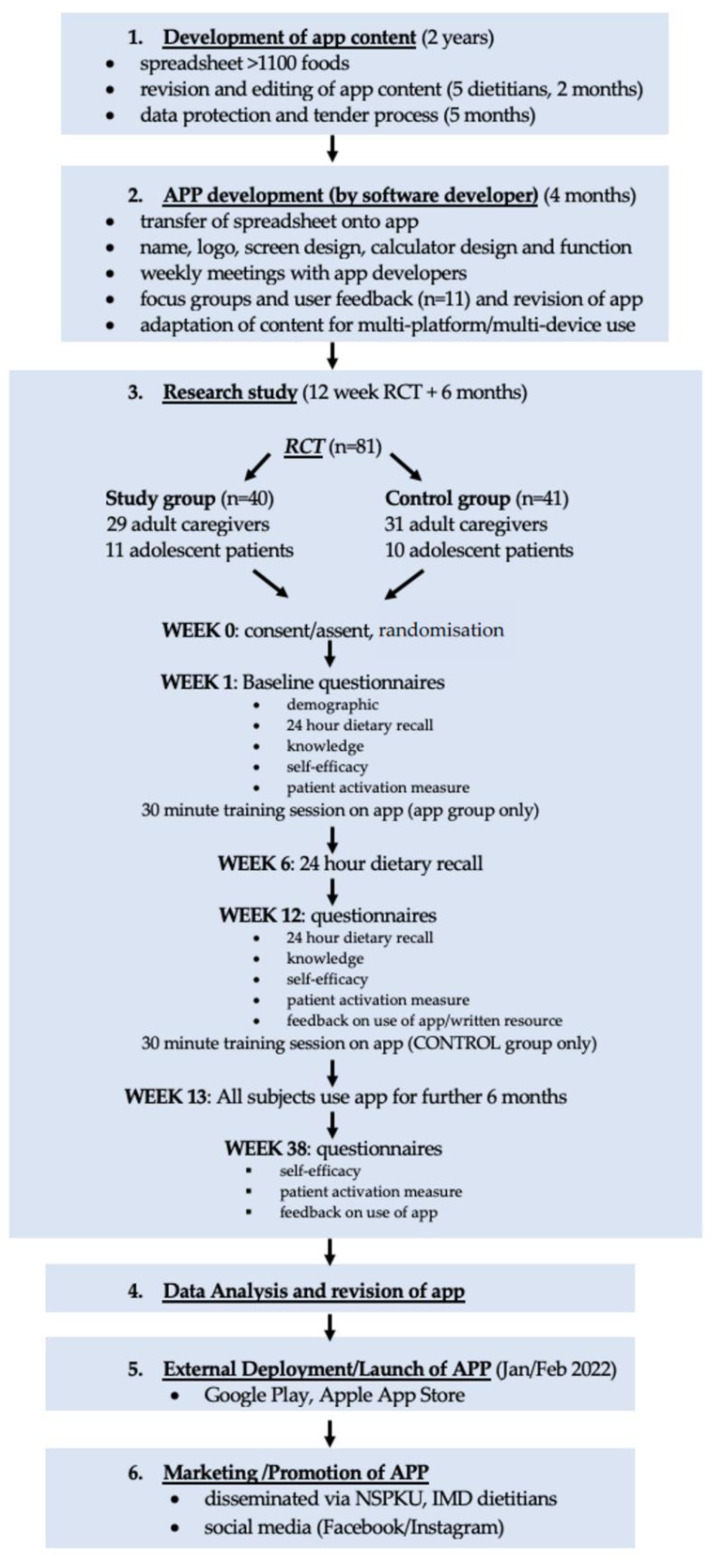
App study design and development procedures. (Abbreviations: RCT = Randomised Controlled Trial; NSPKU = National Society for Phenylketonuria; IMD = Inherited Metabolic Disorder).

**Figure 2 nutrients-14-02182-f002:**
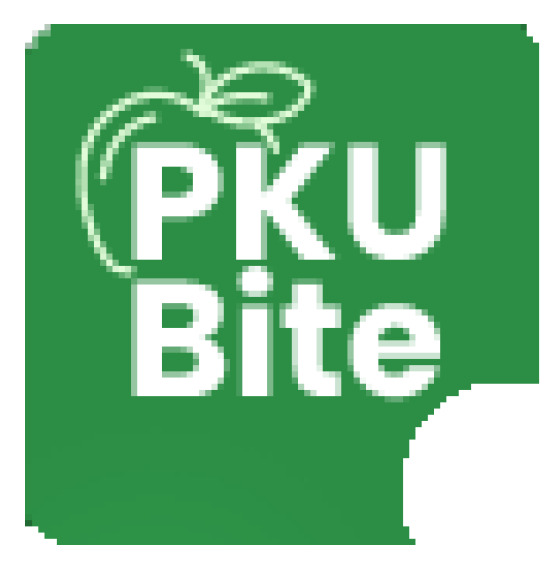
‘*PKU Bite*’^®^ logo.

**Table 1 nutrients-14-02182-t001:** Characteristics of subjects.

	App Group	Control Group
Number of child subjects (gender)	11 (6 F, 5 M)	10 (8 F, 2 M)
Median age of child subjects (years) (IQR)	12.0 (10.5–13.0)	12.7 (12.0–14.0)
Number of parent/carer subjects (gender)	29 (18 mothers, 10 fathers, 1 grandmother)	31 (24 mothers, 5 fathers, 2 grandmothers)
Median age of mothers (years) (IQR)	36.5 (31.8–39.3)	35.5 (31.3–41.0)
Median age of fathers (years) (IQR)	39.0 (35.5–44.5)	38.0 (30.0–44.0)
Mother’s highest educational qualification # median (IQR)	3.0 (2.0–6.5)	2.0 (2.0–3.0)
Father’s highest educational qualification # median (IQR)	5.0 (2.0–6.0)	4.0 (2.0–5.0)
Total number of children (including children of parent/carer participants) (gender)	21 (10 F, 11 M)	25 (15 F, 10 M)
Median age of all children (including children of parent/carer participants) (years) (IQR)	10.0 (5.5–12.5)	9.0 (2.5–12.0)
Ethnicity of all children	19 White/European 1 Asian 1 mixed race	20 White/European 4 Asian 1 mixed race

IQR interquartile range; M male, F female; # Educational levels were recorded as qualification levels 0–8, as described on gov.uk (accessed on 19 April 2018) [22] (0 = no qualifications, 8 = Ph.D.).

**Table 2 nutrients-14-02182-t002:** Baseline use of apps and dietary information.

	Parent/Carer			Child		
Number of Apps Used	App Group *n* = 29 Median (IQR)	Control Group *n* = 31 Median (IQR)	Total *n* = 60 Median (IQR)	App Group *n* = 11 Median (IQR)	Control Group *n* = 10 Median (IQR)	Total *n* = 21 Median (IQR)
Total no. apps used per participant	40 (25–100)	30 (13–61)	36 (18–74)	30 (7–79)	30 (19–47)	30 (10–60)
No. apps used/day per participant	5.5 (3–15)	4 (3–7)	5 (3–10)	4 (1–6)	4.5 (4–6)	4 (3–6)
No. apps used/week per participant	11 (6–25)	8 (6–14)	10 (6–19)	7 (3–12)	7 (4–10)	7 (4–10)
**Sources of dietary information % of subjects (*n*)**						
Written info	86 (25)	81 (25)	83 (50)	36 (4)	60 (6)	48 (10)
Phone/email dietitian	83 (24)	71 (22)	77 (46)	55 (6)	80 (8)	67 (14)
Social media	69 (20)	61 (19)	65 (39)	18 (2)	40 (4)	29 (6)
Clinic/home visit dietitian	55 (16)	39 (12)	47 (28)	45 (5)	30 (3)	38 (8)
Ask others with PKU	21 (6)	23 (7)	22 (13)	9 (1)	10 (1)	10 (2)
Child’s parent(s)	N/A	N/A	N/A	18 (2)	30 (3)	24 (5)

**Table 3 nutrients-14-02182-t003:** Frequency of using app.

	App Group 12 Weeks		App Group 6 Months		Control Group 6 Months	
	Parent (*n* = 29)	Child (*n* = 11)	Parent (*n* = 29)	Child (*n* = 11)	Parent (*n* = 25)	Child (*n* = 7)
Never	5 (2 M, 3 F)	3 (1 M, 2 F)	6 (3 M, 3 F)	3 (1 M, 2 F)	5 (1 M, 4 F)	4 (1 M, 3 F)
Rarely	8 (4 M, 4 F)	4 (1 M, 3 F)	7 (1 M, 6 F)	6 (3 M, 3 F)	5 (1 M, 4 F)	1 (1 F)
Once a week	8 (2 M, 6 F)	2 (1 M, 1 F)	9 (5 M, 4 F)	0	3 (2 M, 1 F)	2 (2 F)
Two–six times per week	4 (2 M, 2 F)	1 (1 M)	7 (1 M, 6 F)	2 (1 M, 1 F)	9 (1 M, 8 F)	0
Once a day	3 (3 F)	1 (1 M)	0	0	1 (1 F)	0
>Once a day	0	0	0	0	2 (2 F)	0
No response	1 (1 F)	0	0	0	0	0

M male, F female.

**Table 4 nutrients-14-02182-t004:** Percentage of respondents accessing different sources of dietary information at least weekly at baseline, 12 weeks, and 6 months.

	Parent/Carer Baseline		Parent/Carer 12 Weeks		Parent/Carer 6 Months		Child Baseline		Child 12 Weeks		Child 6 Months	
	App Group *n* = 29	Control Group *n* = 31	App Group *n* = 29	Control Group *n* = 31	App Group *n* = 29	Control Group *n* = 25	App Group *n* = 11	Control Group *n* = 10	App Group *n* = 11	Control Group *n* = 10	App Group *n* = 11	Control Group *n* = 7
Phone/text/ email dietitian % (*n*)	45 (13)	42 ** (13)	38 (11)	45 (14)	45 (13)	28 ** (8)	9 (1)	50 (5)	0 (0)	30 (3)	18 (2)	43 (3)
*p*-value	0.90		0.41		0.10		0.05		0.01		0.11	
Social media e.g., Facebook/Twitter % (*n*)	72 *^,#^ (21)	65 ^++^ (20)	41 * (12)	55 (17)	52 ^#^ (15)	40 ^++^ (10)	18 (2)	60 (6)	27 (3)	10 (1)	18 (2)	14 (1)
*p*-value	0.54		0.30		0.22		0.45		0.48		0.99	
Written information % (*n*)	34 (10)	71 ^§^ (22)	38 (11)	52 (16)	28 (8)	52 ^§^ (13)	0 (0)	50 (5)	9 (1)	0 (0)	9 (1)	14 (1)
*p*-value	0.004		0.22		0.09		0.06		0.03		0.23	

*p* = Wilcoxon signed rank; * *p* = 0.0003; ^#^ *p* = 0.01; ** *p* = 0.06; ^++^ *p* = 0.001; ^§^ *p* = 0.0009.

**Table 5 nutrients-14-02182-t005:** Self-efficacy and patient activation measure mean scores for app and control groups at baseline, 12 weeks, and 6 months.

	Baseline Median (IQR)	12 Weeks Median (IQR)	6 Months Median (IQR)	*p*-Value * (Baseline vs. 6 Months)
	**Self-efficacy questionnaire (1 = not confident, 10 = confident)**			
App group parent	8.85 (8.6–9.1)	8.70 (8.4–9.1)	9.00 (8.7–9.2)	0.38
Control group parent	8.20 (7.9–8.5)	8.65 (8.2–8.9)	9.20 (9.1–9.3)	0.02
***p*-value ***	**0.02**	**0.70**	**0.11**	
App group child	7.95 (7.7–8.5)	7.60 (6.8–8.4)	8.20 (7.9–8.5)	0.16
Control group child	6.35 (5.6–7.8)	7.05 (6.4–7.5)	6.65 (5.6–7.8)	0.37
***p*-value ***	**0.02**	**0.17**	**0.01**	
App users (parents/carers from app and control groups, who used app ≥weekly)	9.00 (8.3–9.5)	NA	9.10 (8.3–9.8)	0.08
App nonusers (parents/carers from app and control groups, who rarely/never used app)	8.60 (7.7–9.8)	NA	9.45 (8.4–10.0)	0.10
***p*-value ***	**0.75**		**0.51**	
	**Patient activation measure (1 = strongly disagree, 4= strongly agree)**			
App group parent	3.60 (3.4–3.8)	3.60 (3.5–3.8)	3.70 (3.5–3.8)	0.02
Control group parent	3.60 (3.4–3.7)	3.60 (3.4–3.7)	3.70 (3.6–3.8)	0.009
***p*-value ***	**0.93**	**0.39**	**0.42**	
App group child	3.30 (3.1–3.3)	3.10 (3.0–3.2)	3.20 (3.2–3.4)	0.35
Control group child	3.10 (3.0–3.3)	3.20 (2.9–3.3)	2.90 (2.7–3.1)	0.01
***p*-value ***	**0.11**	**0.87**	**<0.0001**	
App users (parents/carers from app and control groups, who used app ≥weekly at 6 months)	3.50 (3.3–3.8)	NA	3.80 (3.5–4.0)	0.03
App nonusers (parents/carers from app and control groups, who rarely/never used app at 6 months)	3.70 (3.2–3.9)	NA	3.50 (3.2–4.0)	0.48
***p*-value ***	**0.76**		**0.24**	

* Wilcoxon signed rank; NA not applicable.

**Table 6 nutrients-14-02182-t006:** Mean percentage of correct answers on protein-labelling knowledge/interpretation questionnaire, by all respondents.

	Baseline Mean% (SD)	12 Weeks Mean% (SD)
App group parent	65.4 (18.0)	66.7 (15.9)
Control group parent	61.4 (14.8)	60.2 (17.3)
App group child	52.1 (22.1)	49.3 (15.3)
Control group child	48.1 (13.7)	47.7 (14.3)
App users (parents/carers from app group, who used app ≥weekly at 12 weeks)	64.9 (18.3)	69.0 (14.1)
App nonusers (parents/carers from app group, who rarely/never used app at 12 weeks)	66.0 (18.4)	63.9 (18.0)

**Table 7 nutrients-14-02182-t007:** Accuracy of calculating protein intake from diet diaries.

		Baseline Mean (SD)	6 Weeks Mean (SD)	12 Weeks Mean (SD)	*p* Value (Baseline vs. 12 Weeks)
App group Parent/carer *n* = 29	% of incorrectly calculated food items *	20.6 (20.9)	15.5 (18.0)	15.5 (18.9)	0.18
	No. of subjects correctly calculating all protein intake	11	9	19	
Control group Parent/carer *n* = 31	% of incorrectly calculated food items *	21.4 (27.4)	24.4 (22.5)	11.5 (19.5)	0.32
	No. of subjects correctly calculating all protein intake	14	7	10	
***p* value (% incorrect app group vs. control group)**		**0.73**	**0.12**	**0.72**	
App group Child *n* = 11	% of incorrectly calculated food items *	28.2 (32.5)	15.9 (20.5)	11.5 (18.2)	0.09
	No. of subjects correctly calculating all protein intake	3	3	4	
Control group Child *n* = 10	% of incorrectly calculated food items *	31.6 (32.8)	40.8 (36.6)	34.3 (39.4)	0.88
	No. of subjects correctly calculating all protein intake	2	0	1	
***p* value (% incorrect app group vs. control group)**		**0.61**	**0.09**	**0.18**	
App users (parents/carers from app group, who used app ≥weekly at 12 weeks) *n* = 16	% of incorrectly calculated food items *	18.2 (21.3)	14.2 (19.2)	16.6 (21.5)	0.97
	No. of subjects correctly calculating all protein intake	6	5	12	
App non-users (parents/carers from app group, who rarely/never used app at 12 weeks) *n* = 13	% of incorrectly calculated food items *	23.7 (29.9)	17.0 (17.0)	14.3 (16.3)	0.13
	No. of subjects correctly calculating all protein intake	5	4	7	
***p* value (% incorrect app users vs. non users)**		**0.54**	**0.61**	**0.89**	

* foods where protein content has not been calculated or has been incorrectly calculated—either over or under the true amount.

**Table 8 nutrients-14-02182-t008:** Median blood Phe levels at baseline, 12 weeks, and 6 months.

	Baseline Median (IQR)	12 Weeks Median (IQR)	6 Months Median (IQR)	*p* Value (Baseline vs. 12 Weeks)	*p* Value (Baseline vs. 6 Months)
App group children *	288 (222–376)	260 (207–397)	315 (232–428)	0.58	0.25
Control group children *	287 (174–338)	221 (198–403)	246 (199–347)	0.59	0.10
***p* value**	**0.50**	**0.52**	**0.22**		

* includes child subjects and children of parent/carer subjects.

**Table 9 nutrients-14-02182-t009:** Most useful features about the app.

	App Group 12 Weeks		App Group 6 Months		Control Group 6 Months	
	Parent % (*n* = 29)	Child % (*n* = 11)	Parent % (*n* = 29)	Child % (*n* = 11)	Parent % (*n* = 25)	Child % (*n* = 7)
Able to look up the number of exchanges in a food	52 (15)	55 (6)	48 (14)	55 (6)	44 (11)	29 (2)
Able to look up protein/Phe content of a food	48 (14)	27 (3)	62 (18)	36 (4)	52 (13)	43 (3)
Able to get advice on suitability of foods	41 (12)	0	52 (15)	18 (2)	52 (13)	0
Quick and easy to access diet information immediately when needed	28 (8)	27 (3)	41 (12)	27 (3)	44 (11)	0
Able to calculate daily exchanges	17 (5)	9 (1)	28 (8)	27 (3)	28 (7)	0
Able to look up guidelines/rules about certain foods	24 (7)	0	28 (8)	18 (2)	20 (5)	0
Able to get advice on how to use foods/ingredients in a low-protein diet	17 (5)	9 (1)	31 (9)	18 (2)	8 (2)	14 (1)
Able to get advice on how to cook/prepare a food	14 (4)	18 (2)	21 (6)	18 (2)	8 (2)	14 (1)
Do not need to contact the dietitian as often	14 (4)	0	14 (4)	18 (2)	16 (4)	0

**Table 10 nutrients-14-02182-t010:** Preferred method of obtaining diet information.

	App Group 6 Months		Control Group 6 Months		Total 6 Months	Direct Quotes
	Parent % (*n* = 29)	Child % (*n* = 11)	Parent % (*n* = 25)	Child % (*n* = 7)	Total % (*n* = 72)	
App	45 (13)	27 (3)	48 (12)	43 (3)	43 (31)	*App preferred because it’s mobile and always with me (much easier than the sheets of paper). App is more discrete* [mother of teenager] *I think it will be really useful when she’s older* [mother of toddler]
Dietitian (phone/email)	38 (11)	18 (2)	52 (13)	71 (5)	43 (31)	*I don’t think the app will ever be able to replace contacting our dietitian who has so much knowledge and experience but it’s definitely very handy and convenient when dietitian is unavailable or when you just want to double check things to be safe* [mother] *I would rather speak to someone re diet* [mother]
Written/pictorial information	28 (8)	9 (1)	28 (7)	14 (1)	24 (17)	*I like the books when they arrive; they make me want to look at them and try new foods. It gives me ideas and my family always try to get me to eat different foods. I tend to stick to the same foods that I like. I don’t like change!* [teenage girl] *I just kept going back to using a calculator and looking in my written paper work for clarity on things, but probably would use the app more if away from home or on holiday* [father of teenager]
Social media	21 (6)	9 (1)	24 (6)	0	18 (13)	*Find that because I am used to using social media that is where I still tend to go back to first* [mother of teenager] *Currently the app does not list branded food items (like on Twitter/written information) so cannot replace those yet* [mother]
Reading food labels	3 (1)	18 (2)	0	0	4 (3)	*It has helped but I have still used other ways of getting info* [mother] *Use a mixture of all* [mother]
Others with PKU	10 (3)	0	8 (2)	0	7 (5)	
Mum	0	18 (2)	4 (1) *	14 (1)	6 (4)	
NSPKU	0	0	4 (1)	0	4 (3)	

Note: some respondents gave more than one choice; * grandparent.

**Table 11 nutrients-14-02182-t011:** Feedback about ‘*PKU Bite*’^®^ app.

App Feedback	Responses	Examples of Direct Quotes
Positive feedback on existing functions: mobile nature of app/useful for shopping like calculator other	5% (*n* = 4) 4% (*n* = 3) 2% (*n* = 2)	*Liked green/red/orange colours-found this very helpful* [mother] *I loved the portion size option on the protein exchange calculator. I used this the most* [mother] *Very easy to use; calculator easier than the card* [mother] *Easier to use app when shopping* [mother] *It was useful out and about not carrying the food booklets* [teenage boy] *Mobile and quick* [grandmother]
Difficulties with app: finding foods want more branded items using calculator	9% (*n* = 7) 11% (*n* = 9) 4% (*n* = 3)	*Sometimes I would search up a food and it wouldn’t come up, but when I went through the categories I found it* [teenage girl] *More information on everyday foods e.g., crisps, more supermarket foods* [mother] *Would like to see foods included like ‘McDonalds’ and other restaurants* [mother] *Calculator was confusing at first but got used to it* [mother of teenager]
Reasons for not using the app: not into technology set in ways/unchanging diet	2% (*n* = 2) 9% (*n* = 7)	*I am very stuck in my ways and not great with technology* [father of teenager] *I am used to calculating diet and my son does not change much what he eats* [mother of teenager] *I know what to eat and I don’t change what I eat* [teenage boy] *I can imagine it would be better for new families; I can definitely see it would help people* [teenage boy]
Ideas for additional functions: Exchange tracker Label scanner More recipes Other ideas	11% (*n* = 9) 9% (*n* = 7) 7% (*n* = 6) 10% (*n* = 8)	*Meal tracker function. Multiple users log in to the same account-sharing information across family members* [mother] *Food diary-to be able to record how many exchanges he had and from what food* [mother] *Section to enter your exchanges and medicine so you can have everything in one place. Also access for the dietitian to see* [father] *Ability to scan the barcode of a food packet and it tells you exchanges/100 g and exchanges/portion* [teenage boy] *More meal ideas for toddlers; easy quick meals* [mother] *Recommendations and ideas for meals and recipes that take into account exchanges* [father of teenager] *Digital versions of the picture books-I usually take pictures on my phone of the picture books* [mother] *Would like index of recipes so know what recipes are on the app* [father of teenager]

## Data Availability

Not applicable.

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
