# Peer review of "Efficacy of a New Low-Protein Multimedia Diet App for PKU"

_nutrients, 2022, doi:10.3390/nu14112182_

Round 1
Reviewer 1 Report
The first dietary mobile application "PKU Bite" for PKU patients is comprehensively presented in the manuscript. I have only two comments to the Authors and I am asking for clarification:
(1) 12 food categories were distinguished. One of them includes fats, oils, dips and spices. What impact on the education of patients and their caregivers has the combination of refined oils and ketchups, margarines and dressings with ground seeds rich in phenylalanine? In the discussion part of the manuscript, the role of the app in the selection of food products for meals as well as in general nutritional education is emphesized.
(2) How numerous was the group of partcipants who were very unsatisfied with the app and how many simply refused to answer?
Author Response
REVIEWER 1
The first dietary mobile application "PKU Bite" for PKU patients is comprehensively presented in the manuscript. I have only two comments to the Authors and I am asking for clarification:
(1) 12 food categories were distinguished. One of them includes fats, oils, dips and spices. What impact on the education of patients and their caregivers has the combination of refined oils and ketchups, margarines and dressings with ground seeds rich in phenylalanine? In the discussion part of the manuscript, the role of the app in the selection of food products for meals as well as in general nutritional education is emphasized.
Thank you. The categories were grouped subjectively for convenience and food similarity/purpose rather than food suitability in the diet. It iis important that people with PKU or their caregivers have information about all foods. The food categories are another method of searching for a particular food instead of the ‘search by food name’. All food categories contain foods that should be calculated/measured as an exchange food or foods that are unsuitable. The app is based on rules according to the British Inherited Metabolic Diseases Group (BIMDG) dietitians National Consensus Statements on Food Labelling Interpretation an Protein Allocation in a Low Phenylalanine Diet. In these statements it was agreed that for most products containing protein ≤0.5g/100g, they could be consumed without calculating the protein, if they contain >0.5g/100g they must be calculated as part of the daily protein intake. This consensus paper issues advice on foods that contain seeds as an ingredient. The app gives advice about the suitability of individual food, so the education is provided about individual foods not food categories.
(2) How numerous was the group of participants who were very unsatisfied with the app and how many simply refused to answer?
Only 2/72 (3%) participants reported being very unsatisfied with the app (1 parent and 1 adolescent) and 14/72 (19%) did not respond. This is reported in the paper line 433.

Reviewer 2 Report
Line 98: would suggest (…via any smart phone or tablet)
- Would like to see more than 1 center included in trial to ensure app is effective despite differences in level of teaching given by clinical team and care protocols.
- Study design seems to include as many self-reported methods as possible to assess efficacy of the app.
- Suggest app would be most effective if use is self-guided and does not require a 30 min training session
- Would like to have included in the discussion why the authors think use of the app went up with parents and down with children?
- Disappointing that knowledge and interpretation of low protein labelling didn’t improve statistically nor did phe concentrations – although it is noted that the median levels were within control range. Also disappointing that app did not statistically improve ability to adhere to diet with more accurate protein counting.
- Interesting that a possible use of the app would be to lessen the tendency to unnecessary strictness in some parents.
- Good and thorough discussion for why some groups did not find the app as useful as others did.
- I think it is important to include in the article the parameters for including or not including a particular food and from what source nutrient values were derived.
Author Response
REVIEWER 2
Line 98: would suggest (…via any smart phone or tablet)
Thank you, the word “smart” has been added as suggested (now line 99).
- Would like to see more than 1 center included in trial to ensure app is effective despite differences in level of teaching given by clinical team and care protocols.
Thank you, we have added a comment about this in the discussion and a recommendation that this is necessary research in the future (line 588-92).
- Study design seems to include as many self-reported methods as possible to assess efficacy of the app.
Thank you
- Suggest app would be most effective if use is self-guided and does not require a 30 min training session
The 30 minute training session was only done for the purpose of the study. Some of this time was spent in explaining how to download the trial version of the app onto their device. This has been clarified in the methods (line 188-9). The app is now available on the App store and on Google Play and can be downloaded easily and also includes a brief set of instructions on how to use the app. A statement about this is also included in the discussion (line 525).
- Would like to have included in the discussion why the authors think use of the app went up with parents and down with children?
Use of the app did not go down dramatically with children. In Table 3 only 2 children used the app less at 6 months compared to 12 weeks. Small subject numbers make this difficult to interpret.
However, the children included in the study were teenagers, a group who are known to be difficult to engage and who are well accustomed with a low phenylalanine diet, and have coped with their diets for >10 years. They also tend to not to alter their food choices, varying little from day to day in the types of foods that they eat and are reluctant to change or try new foods. They likely used the app initially out of curiosity but quickly lost interest. As one teenager noted, “I can imagine it would be useful for new families”. Teenagers and parents of teenagers commonly noted that they were used to calculating protein in their diet and they knew the protein content of the foods they commonly ate. This is covered in the discussion (lines 474-495) and we have added an extra line to clarify (line 481-3).
- Disappointing that knowledge and interpretation of low protein labelling didn’t improve statistically nor did phe concentrations – although it is noted that the median levels were within control range. Also disappointing that app did not statistically improve ability to adhere to diet with more accurate protein counting.
Thank you for your comments. It was disappointing that results were not statistically significant, but as stated in the paper, these were patients who were already well established on their diet and well educated in a Phe restricted diet. Results with a group of parents with much younger children may show more statistically significant results. In addition, people vary in their learning styles and in the ways they prefer to receive information. Technology and apps are not for everyone, so it is not surprising that only a percentage of the participants regularly used the app and those who did not, undoubtedly diluted any benefits the app may have instilled in those who did.
- Interesting that a possible use of the app would be to lessen the tendency to unnecessary strictness in some parents.
Yes, as the study mentioned in lines 512-516, we have found that parents and patients are frequently stricter in their interpretation of a low phenylalanine diet than dietitians. Some of this is due to uncertainty about the suitability of some foods.So, for some, the app may reassure them that they can include foods they previously may have avoided.
- Good and thorough discussion for why some groups did not find the app as useful as others did.
Thank you
- I think it is important to include in the article the parameters for including or not including a particular food and from what source nutrient values were derived.
All foods or food groups that might be consumed on a phenylalanine restricted diet are included in the app (this has been clarified in the manuscript (lines 86-7)). The information on food suitability in the app is a largely based on interpreting food labels, therefore, no nutrient values are provided as this will vary from brand to brand. The parameters for including or not including a food in the diet is based on the British Inherited Metabolic Diseases Group (BIMDG) dietitians National Consensus Statements on Food Labelling Interpretation an Protein Allocation in a Low Phenylalanine Diet. This is referenced in the paper (and on the app under ‘medical references’ in the main menu).This information is too lengthy to include in the paper but can be accessed in full via the reference.
